# Association between surgeon training grade and the risk of revision following total knee replacement: An analysis of National Joint Registry data

Timothy J. Fowler[1]*, Nicholas R. Howells[1], Ashley W. Blom[2,3], Adrian Sayers[1,2‡], Michael R. Whitehouse[1,2‡]

1 Musculoskeletal Research Unit, Translational Health Sciences, Bristol Medical School, Southmead Hospital, Bristol, United Kingdom, 2 National Institute for Health Research Bristol Biomedical Research Centre, University Hospitals Bristol NHS Foundation Trust, University of Bristol, Bristol, United Kingdom, 3 Faculty of Biology, Medicine and Health, University of Manchester, Manchester, United Kingdom

‡ These authors are joint senior authors on this work.
* tfowler@doctors.org.uk

## Abstract

### Background

Total knee replacements (TKRs) are performed by surgeons at different stages in training with varying levels of supervision, but we do not know if this is a safe practice or whether trainees achieve equivalent outcomes to consultant-performed TKR. This study aimed to investigate the association between surgeon grade, the supervision of trainees, and the risk of revision following TKR. Revision is defined by the National Joint Registry (NJR) for England and Wales as any procedure to add, remove, or modify one or more components of an implant construct for any reason.

### Methods and findings

We conducted an observational study using prospectively collected data from the NJR. We included 953,081 cases in 788,288 adult patients who underwent primary TKR for osteoarthritis (OA), recorded in the NJR between 2003 and 2019. Exposures were surgeon grade (consultant or trainee) and the level of scrubbed consultant supervision of trainees. The primary outcome was all-cause revision, and the secondary outcome was the number of procedures revised for the following indications: aseptic loosening/lysis, infection, progression of OA, unexplained pain, and instability. Flexible parametric survival models (FPM) were incrementally adjusted in the following manner. Model 1 was unadjusted. Model 2 was adjusted for patient-level factors (age, sex, American Society of Anaesthesiologists [ASA] grade, and index of multiple deprivation [IMD] decile). Model 3 was further adjusted for operation-level factors (anaesthetic, approach, fixation, constraint and whether or not the patella

**Data availability statement:** The data used in the study are available from The National Joint Registry (NJR) (https://www.njrcentre.org.uk). Restrictions apply to the availability of these data, which were used under license for the current study, and are therefore not publicly available. Data access applications can be made to the NJR Research Committee. With NJR permission in place, the data underlying the presented results will be available to access via the NJR data access network. The authors of this manuscript are not the data owner and do not have permission to share the data. The code used in the analysis is available from Github [https://github.com/tjfowler89/Stata_Code_Fowler/tree/main] and archived in Zenodo [https://doi.org/10.5281/zenodo.15785216].

**Funding:** Posts of members of the research team were funded by a contract grant from the National Joint Registry, in the form of the Lot 2 contract (FTS 010307-2022: Statistical Analysis, Support and Associated Services—MRW, AWB and AS). This study was also supported by the National Institute for Health Research (NIHR) Biomedical Research Centre at the University Hospitals Bristol NHS Foundation Trust and the University of Bristol (IS-BRC-1215-20011—MRW and AWB). TF was supported by a NIHR Academic Clinical Fellowship. AS was supported by an MRC strategic skills fellowship (MR/L01226X/1). The views expressed in this publication are those of the authors and not necessarily those of the NHS, the NIHR, or the Department of Health and Social Care. The funders had no role in study design, data collection and analysis, the preparation of the manuscript, or the decision to publish.

**Competing interests:** We have read the journal's policy and the authors of this manuscript have the following competing interests: AWB, MRW and AS report holding a contract with The Healthcare Quality Improvement Partnership/The National Joint Registry in the form of the Lot 2 contract (FTS 010307-2022: Statistical Analysis, Support and Associated Services), during the conduct of the submitted work. MRW and AWB were supported by the NIHR Biomedical Research Centre at University Hospitals Bristol and Weston NHS Foundation Trust and the University of Bristol (IS-BRC-1215-20011), during the conduct

was resurfaced). Model 4 was further adjusted for healthcare setting factors (funding source, and year of operation). Trainees performed 96,544 (10.1%) TKRs and were directly supervised by a scrubbed consultant in 63.2% of trainee-performed cases. Trainees achieved comparable outcomes to consultants in terms of the unadjusted cumulative probability of all-cause revision (e.g., 15 years of follow-up: consultant % Failure 4.79 (95% CI [4.67, 4.92]) versus trainee (overall) % Failure 4.75 (95% CI [4.43, 5.10]). Adjusted FPM analysis indicated evidence of an association between trainee-performed TKR and a small increased risk of early all-cause revision up to, but not exceeding, 4 years follow-up (1 year: HR 1.12 (95% CI [1.05, 1.19]), 4 years: HR 1.00 (95% CI [0.95, 1.06]), 16 years: HR 0.89 (95% CI [0.81, 0.98])). This association was not explained by the level of supervision. Further analysis suggested that this association may be attributable to revisions for aseptic loosening/lysis, infection, and progression of OA (i.e., subsequent patellar resurfacing). Limitations of this study relate to its observational design and include: the potential for non-random allocation of cases by consultants to trainees; residual confounding; and the use of the binary variable 'surgeon grade', which does not capture variations in the level of experience between trainees.

## Conclusions

Trainees in England and Wales achieve safe and acceptable all-cause TKR implant survival, with comparable outcomes to consultants. However, adjusted analyses suggest an association between trainee-performed TKR and a small increase in the risk of early all-cause revision. This association may be attributable to factors including aseptic loosening, infection, and progression of OA. Current training practices for TKR in England and Wales are safe in terms of equivalence of all-cause implant survival to consultant-performed TKR, but we have identified areas for potential improvement in trainee outcomes.

## Author summary

### Why was this study done?

- Total knee replacements (TKRs) are performed by surgeons at different stages of training, but it is unclear whether outcomes differ between procedures performed by trainees and those performed by fully trained consultants.

- Smaller studies have suggested that trainees achieve acceptable outcomes, but there was limited large-scale evidence on long-term implant survival and revision risk.

- This study aimed to evaluate the risk of revision following trainee-performed TKR using a national registry to inform training practices and patient care.

of the submitted work. AWB and MRW report grants from the NIHR investigating the outcomes of joint replacement, outside the submitted work; AWB and MRW are editors of an Orthopaedic textbook for which they receive royalty payments from Taylor Francis. MRW conducts teaching on courses sponsored by Heraeus and DePuy for which his institution receives market rate payments. All other authors declare no conflicts of interest.

**Abbreviations:** ASA, American Society of Anaesthesiologists; BMI, body mass index; CCT, Certification of Completion of Training; F1, Foundation Year 1; FPM, flexible parametric survival modelling; HR, hazard ratio; IMD, index of multiple deprivation; IQR, interquartile range; KM, Kaplan–Meier; NJR, National Joint Registry; NZJR, New Zealand Joint Registry; OA, osteoarthritis; PH, proportional hazards; PROMS, patient-reported outcome measures; SD, standard deviation; ST2, Specialty Trainee Year 2; TKRs, total knee replacements.

## What did the researchers do and find?

- We analysed 953,081 primary TKRs recorded in the National Joint Registry for England and Wales between 2003 and 2019.

- We compared the risk of revision between trainee- and consultant-performed TKRs, adjusting for patient, surgical, and healthcare factors. We used a statistical method called 'flexible parametric survival modelling' to analyse the data.

- We found that TKRs performed by trainees had a small increased risk of early revision within the first four years, but long-term outcomes were comparable, and implant survival remained within recognised safety benchmarks.

## What do these findings mean?

- Trainees in England and Wales achieve safe and acceptable all-cause TKR implant survival, with comparable outcomes to consultants.

- The small increased risk of early revision following trainee-performed procedures may relate to factors such as infection, implant loosening, and progression of osteoarthritis, suggesting areas where surgical training can be improved.

- The findings are reassuring and support the current methods by which surgeons are trained to perform TKR in England and Wales, with an emphasis on supervision and targeted support during critical steps of the procedure to improve trainee outcomes.

- Limitations of this study relate to its observational design. A notable limitation is the use of a binary exposure (consultant, or trainee), which does not capture variations in the level of experience between trainees.

### Introduction

Total knee replacement (TKR) is a clinically and cost-effective treatment for end-stage osteoarthritis (OA). It is one of the most common elective surgical procedures worldwide with over 100,000 performed annually in the United Kingdom (UK), and over 700,000 performed annually in the United States of America (USA) [1]. TKRs are performed by surgeons at different stages in training, with varying levels of supervision. There is a balance between ensuring that trainees get adequate operative experience to develop the skills required for independent consultant practice, while ensuring safe and acceptable outcomes for patients.

There are challenges when quantifying the safety and efficacy of surgical interventions, but objective clinical outcomes can be useful metrics of success. The revision-free survival of a joint replacement is a commonly used objective clinical outcome measure. Our current understanding of the survival of TKRs in the context of surgical training is limited.

Studies of TKR outcomes have shown that trainees can achieve comparable results to consultants in terms of implant alignment [2–4], patient-reported outcome

measures (PROMs) [4–7], and complication rates [8,9]. Our recent meta-analysis, which included 936 TKRs, found no strong evidence of an association between surgeon grade and the net survival of TKRs at 10 years [10]. Furthermore, New Zealand Joint Registry (NZJR) evidence has suggested that implant survival is not compromised in trainee-performed TKR [7]. However, this study did not investigate the indication for revision, and limitations in the statistical modelling employed leave scope for further investigation.

The aim of this study was to use National Joint Registry (NJR) data for England and Wales, to investigate the association between surgeon grade (consultant or trainee), the supervision of trainees, and the risk of revision following TKR. An additional aim of this study was to investigate the association between surgeon grade and the risk of revision for the following indications: aseptic loosening/lysis, infection, progression of OA, unexplained pain, and instability.

## Methods

### Patients and data sources

We performed an observational study using prospectively collected data recorded in the NJR. The base dataset was 1,502,564 linked knee procedures recorded in the NJR between 1 April 2003 and 31 December 2019. We included primary TKRs in adult patients (≥18 years) performed in England and Wales for an indication of OA only. The analysis was restricted to cases performed for an indication of OA to facilitate relative standardisation of procedural complexity and limit confounding by indication. The study end date was limited to predate the anomalous period of elective orthopaedic practice during the COVID-19 pandemic [11]. Patients were consented for inclusion in the NJR according to standard practice [12].

Cases were included if the operating surgeon grade was recorded as any of the following: Foundation Year 1 (F1) to Specialty Trainee Year 2 (ST2); ST3-ST8; fellow; or consultant. F1-ST2 is the most junior training category recorded in the NJR, followed by ST3-ST8. The process of mapping grade classifications to account for variations in terminology used in different versions of the Minimum Data Set (MDS) form is detailed in S1 Appendix.

### Data access and processing

The base dataset used in the current study is based on the same cut of NJR data that is used in the 17th Annual Report [13]. NJR data are annually linked to other healthcare system datasets, including the Hospital Episodes Statistics (HES) service, and Civil Registration Authority data, using unique patient identifiers. This linkage, which was carried out by the NJR prior to us obtaining the dataset, is approved by the Health Research Authority under Section 251 of the NHS act 2006 [13]. NJR data completeness is routinely monitored using a published data quality audit process [14]. The steps taken in data processing and are summarised in Fig 1 and illustrated in greater detail in S1 Fig. All exclusions are consistent with the exclusion criteria of this study and the stage at which these occurred is clearly documented. Complete-case analysis was used in all analyses and records with missing data in any confounding variable field used in subsequent statistical models were excluded from the relevant model. Details of missing data, the number of cases excluded, and the reasons for exclusion are documented in S2 Fig.

### Exposures

The primary exposure (exposure A) was surgeon grade, which was categorised into two groups according to the grade of the operating surgeon: (1) consultant, or (2) trainee. Procedures performed by surgeons of the following grades were categorised under the variable 'trainee': F1-ST2; ST3-ST8; and fellow. Consultant surgeons have been awarded a Certificate of Completion of Training (CCT) in orthopaedic surgery and have been appointed to a senior position in which they can supervise trainees. The term 'consultant' is synonymous with 'attending' in several healthcare settings, including the USA.

F1-ST2 represents the first four years of postgraduate training after graduating from medical school (F1, F2, ST1 and ST2). ST2 doctors who have completed the Membership of the Royal College of Surgeons (MRCS) examination are

PLOS Medicine

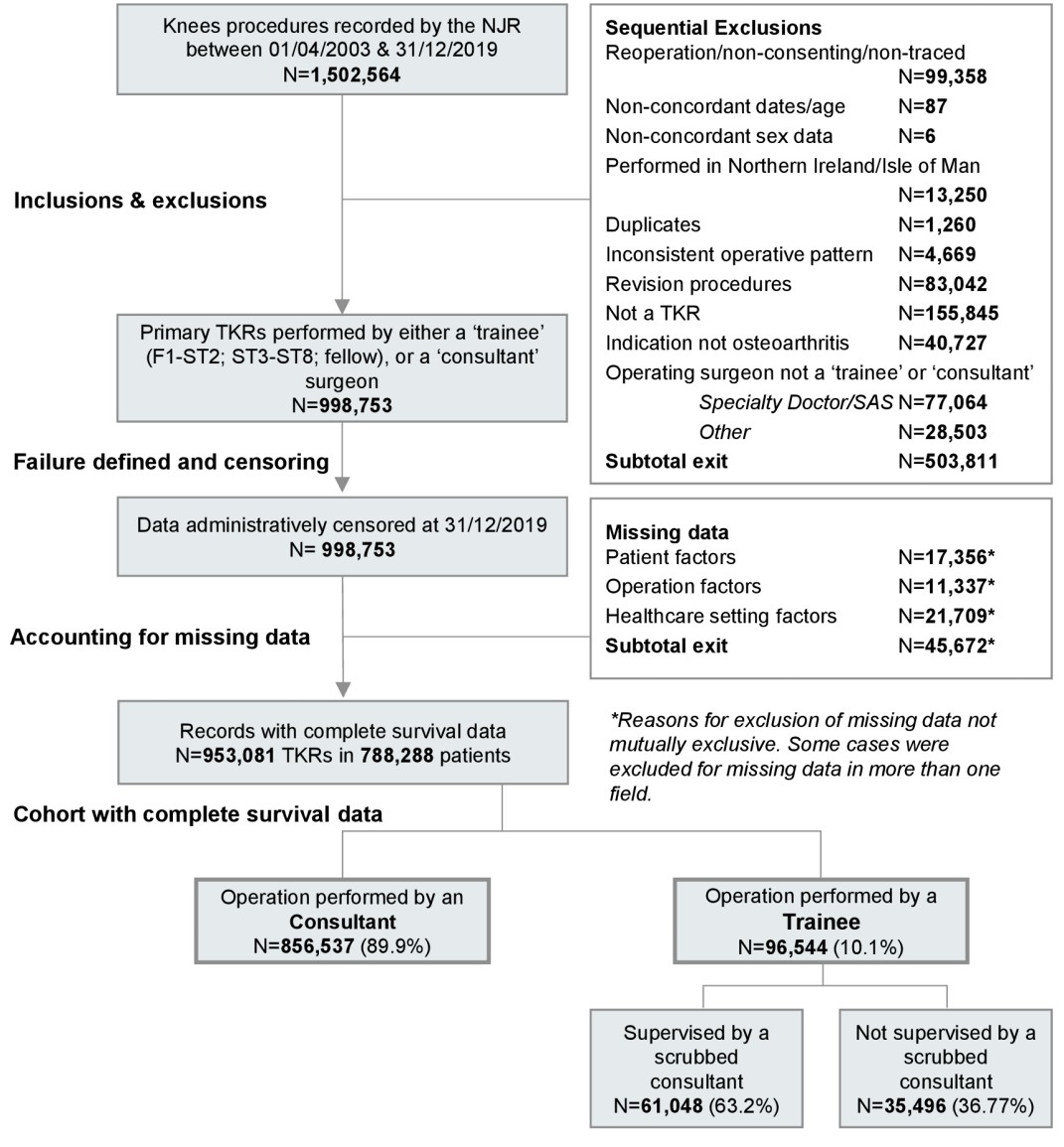

TKR, total knee replacement; NJR, National Joint Registry; F1, foundation year 1 doctor; ST, specialty trainee (number denotes year); SAS, associate specialist surgeon.

**Fig 1. Study flow diagram.**

eligible to apply to Specialty Training. Specialty Training in Trauma and Orthopaedic Surgery is typically a six-year programme (ST3–8). ST3-ST8 trainees are referred to as 'specialty trainees', or 'registrars'. Progression through training levels is dependent on the successful completion of training requirements and competencies. Trainees who have completed ST6 are eligible to sit the examination for Fellowship of the Royal College of Surgeons (FRCS), which is mandatory for Certification of Completion of Training (CCT). Trainees subsequently progress to post-CCT fellowship training prior to applying for a consultant position. The term 'consultant' is synonymous with 'attending', and the term 'registrar' is

synonymous with 'resident' in many healthcare settings, including the USA. A schematic summary of the stages of surgical training in the UK is included in S2 Appendix [15].

The secondary exposure was whether or not trainees were directly supervised by a scrubbed consultant during the procedure (exposure B). Trainee cases were subcategorised accordingly: (1) trainee supervised by a scrubbed consultant, or (2) trainee not supervised by a scrubbed consultant. Cases were categorised as 'supervised by a scrubbed consultant' if the first assistant was recorded as a consultant.

To account for variations in the level of experience between surgical trainees, we performed an additional analysis in which procedures were categorised according to the specific training grade of the operating surgeon (exposure C: consultant, F1-ST2, ST3-ST8, or fellow).

## Outcomes of interest

The primary outcome measure was all-cause revision, which is defined by the NJR as any procedure to add, remove, or modify one or more components of an implant construct for any reason [11]. The secondary outcome measure was the number of primary procedures revised for specific indications. We considered the following indications, which are listed as the five most common indications for knee replacement revision by the NJR: aseptic loosening/lysis, infection, progression of OA, unexplained pain, and instability [13]. Each indication for revision was examined as a separate survival endpoint and other events were administratively censored.

## Statistical analysis

Categorical variables were described with frequencies and percentages. Continuous variables were described with the mean, standard deviation (SD), and interquartile range (IQR). Records were either administratively censored on 31 December 2019, or the date of death, whichever was earliest. Estimates of unadjusted net implant failure were calculated using Kaplan–Meier (KM) analysis.

We performed an initial analysis using Cox regression. A combination of Schoenfeld residual plots/tests and likelihood ratio testing (comparing proportional and non-proportional hazards models) was used to assess the proportional hazards (PH) assumption at each level of adjustment and assess the time-dependent effects of variables [16]. Neither crude nor adjusted analyses satisfied the PH assumption, which was explained by the time-dependent effect of surgeon grade (exposure A).

To account for non-proportionality, we used flexible parametric survival modelling (FPM), which uses restricted cubic spline functions to model the baseline hazard and account for the time-dependent effects of specified variables [16,17]. We used graphical assessment, AIC and BIC, and likelihood ratio testing to select the most parsimonious model [16]. The process of model selection, construction and justification is explained in greater depth in S3 Appendix.

The analyses were incrementally adjusted for the following categorical confounding variables. Model 1 was unadjusted. Model 2 was adjusted for patient-level factors (age, sex, American Society of Anaesthesiologists [ASA] grade, and index of multiple deprivation [IMD] decile). Model 3 was further adjusted for operation-level factors (anaesthetic, approach, fixation, constraint and whether or not the patella was resurfaced). Model 4 was further adjusted for healthcare setting factors (funding source, and year of operation). In each case, the baseline category was the most frequently occurring (as detailed in S4 Appendix).

Body mass index (BMI) is missing in a large proportion of NJR records. It has been reported that approximately 40% of patients did not have a BMI recorded in the NJR in 2009, compared to approximately 18% in 2016 [18]. Due to the significant proportion of records with missing values, BMI was not included as a confounding variable in the primary analyses. This is consistent with the approach used in previous NJR studies and this decision was made prior to initiating the study based on the known pattern of missing data [19,20]. However, in response to the peer review process, BMI has been included in a sensitivity analysis limited to cases with complete BMI data. An additional sensitivity analysis was introduced in response to the peer-reviewed process, in which the analysis was restricted to NHS-funded cases performed in NHS hospitals, thereby reducing healthcare setting-related confounding.

A further sensitivity analysis was performed using KM to investigate the influence of surgeon grade on the cumulative probability of failure, compared to other operative factors, including fixation, patellar resurfacing, and constraint. All analyses were performed using Stata (Version SE 15.1; StataCorp LP, USA). This study is reported as per the Reporting of studies Conducted using Observational Routinely-collected health Data (RECORD) Statement (S1 RECORD Checklist) [21].

### Ethics statement

The NJR supports public health surveillance and wider clinical decision-making and holds data that are anonymous to the researchers who use it. NHS Health Research Authority guidance dictates that the secondary use of such data for research does not require approval by a research ethics committee. Therefore, separate research ethics committee approval was not required for this study. Patients are consented for inclusion in the NJR according to standard practice, with permission under the Health Service (Control of Patient Information) Regulations, otherwise referred to as Section 251 support [22].

## Results

### Descriptive analysis

We included 953,081 TKRs in 788,288 patients, with a maximum duration of follow-up of 16.8 years. Trainees performed 96,544 TKRs (10.1%) and were supervised by a scrubbed consultant in 61,048 (63.2%) of these cases (Fig 1). Mean follow-up was 6.3 years (SD 3.9; IQR 3.0 to 9.2 years) for trainee-performed TKRs and 5.8 years (SD 3.9; IQR 2.5 to 8.5 years) for consultant-performed TKRs. A total of 21,572 knees were revised at a mean of 3.4 years (SD 3.0; IQR 1.3 to 4.7 years).

Demographic data and summary statistics for confounding variables are listed in Table 1. The mean age of patients operated on by trainees was 1.2 years older than patients operated on by consultants (70.6 versus 69.4 years). Trainees operated on a lower proportion of ASA I patients (7.9% versus 10.6%) and a higher proportion of ASA≥III patients (21.7% versus 16.3%) compared to consultants. A higher proportion of trainee procedures utilised cemented implants (96.2% versus 95.1%) and a lower proportion of trainee-performed cases included patellar resurfacing compared to consultant cases (34.5% versus 38.9%). Trainees have performed a lower proportion of cases since 2012 (9.1% since 2012 versus 12.0% pre-2012) and a higher proportion of trainee procedures have been supervised by a scrubbed consultant since 2012 (64.0% since 2012 versus 36.0% pre-2012).

### All-cause revision: Unadjusted Kaplan–Meier (KM) analysis

The unadjusted cumulative probability of all-cause revision according to surgeon grade (exposure A) and supervision (exposure B) is documented in Table 2 and displayed as a KM plot (one minus survival) in Fig 2. The Orthopaedic Data Evaluation Panel (ODEP) A* thresholds are presented alongside for comparison to an internationally recognised benchmark for implant component longevity.

The KM plot in Fig 2A shows that the cumulative probability of failure of trainee-performed TKRs follows a very similar trend to that of consultant-performed TKRs. There is a subtle divergence in the probability of failure between 1 and 4 years, with separation between the confidence intervals noted at the 3-year interval. However, the confidence intervals for consultant- and trainee-performed TKRs overlap at all other intervals of follow-up (Table 2). The upper confidence interval for the cumulative probability of failure of trainee-performed TKRs is below the ODEP A* threshold at all intervals of follow-up.

The KM plot in Fig 2B shows that the cumulative probability of failure of trainee-performed TKRs follows a similar trend, regardless of the level of scrubbed consultant supervision (exposure B). The upper confidence interval of both plots is below the ODEP A* threshold at all intervals of follow-up. The confidence intervals overlap at all time points (Fig 2B and Table 2), which suggests that there is no difference in the cumulative probability of failure between the two trainee groups.

**Table 1. Descriptive statistics for patient, operation, and healthcare setting factors for included TKRs.**

| Variable | Surgeon grade and supervision (n=953,081) | | | |
| --- | --- | --- | --- | --- |
| | Consultant (*n*=856,537) | Trainee (overall) (*n*=96,544) | Trainee supervised by scrubbed consultant (*n*=61,048) | Trainee not supervised by scrubbed consultant (*n*=35,496) |
| **Mean age (SD) [years]** | 69.4 (9.2) | 70.6 (8.9) | 70.5 (8.9) | 70.8 (8.7) |
| **Age groups (%)** | | | | |
| <55 | 51,856 (6.1) | 4,001 (4.1) | 2,654 (4.4) | 1,347 (3.8) |
| 55–64 | 198,108 (23.1) | 19,667 (20.4) | 12,617 (20.7) | 7,050 (19.9) |
| 65–74 | 339,617 (39.7) | 38,671 (40.1) | 24,349 (39.9) | 14,322 (40.4) |
| 75–84 | 232,999 (27.2) | 29,637 (30.7) | 18,581 (30.4) | 11,056 (31.2) |
| >85 | 33,957 (4.0) | 4,568 (4.7) | 2,847 (4.7) | 1,721 (4.9) |
| **Female (%)** | 488,057 (57.0) | 55,688 (57.7) | 35,082 (57.5) | 20,606 (58.1) |
| **Side (% right)** | 451,738 (52.7) | 50,582 (52.4) | 31,961 (52.4) | 18,621 (52.5) |
| **IMD decile (%)** | | | | |
| 1–2 (most deprived) | 121,773 (14.2) | 15,709 (16.3) | 10,497 (17.2) | 5,212 (14.7) |
| 3–4 | 150,768 (17.6) | 18,198 (18.9) | 11,397 (18.7) | 6,801 (19.2) |
| 5–6 | 186,143 (21.7) | 21,167 (21.9) | 13,226 (21.7) | 7,941 (22.4) |
| 7–8 | 199,725 (23.3) | 21,309 (22.1) | 13,220 (21.7) | 8,089 (22.8) |
| 9–10 (least deprived) | 198,128 (23.1) | 20,161 (20.9) | 12,708 (20.8) | 7,453 (21.0) |
| **BMI (kg/m²)** | | | | |
| <19 (underweight) | 1,332 (0.2) | 126 (0.1) | 81 (0.1) | 45 (0.1) |
| 19–24.9 (normal) | 57,186 (6.7) | 5,657 (5.9) | 3,620 (5.9) | 2,037 (5.7) |
| 25–29.9 (overweight) | 197,325 (23.0) | 19,836 (20.6) | 13,172 (21.6) | 6,664 (18.8) |
| >30 (obese) | 332,930 (38.9) | 36,095 (37.4) | 24,077 (39.4) | 12,018 (33.9) |
| Missing | 267,764 (31.3) | 34,830 (36.1) | 20,098 (32.9) | 14,732 (41.5) |
| **ASA grade (%)** | | | | |
| ASA I | 91,060 (10.6) | 7,644 (7.9) | 4,831 (7.9) | 2,813 (7.9) |
| ASA II | 625,871 (73.1) | 67,980 (70.4) | 42,506 (69.6) | 25,474 (71.8) |
| ASA ≥III | 139,606 (16.3) | 20,920 (21.7) | 13,711 (22.5) | 7,209 (20.3) |
| **Anaesthetic (%)** | | | | |
| Spinal | 580,362 (67.8) | 62,254 (64.5) | 41,480 (68.0) | 20,774 (58.5) |
| General | 312,453 (36.5) | 35,447 (36.7) | 20,198 (33.1) | 15,249 (43.0) |
| Epidural | 46,068 (5.4) | 7,242 (7.5) | 4,143 (6.8) | 3,099 (8.7) |
| Nerve block | 122,671 (14.3) | 17,145 (17.8) | 9,606 (15.7) | 7,539 (21.2) |
| **Approach (%)** | | | | |
| Lateral parapatellar | 7,301 (0.9) | 689 (0.7) | 343 (0.6) | 346 (1.0) |
| Medial parapatellar | 801,543 (93.6) | 90,145 (93.4) | 57,009 (93.4) | 33,136 (93.4) |
| Mid-vastus | 21,446 (2.5) | 2,482 (2.6) | 1,705 (2.8) | 777 (2.2) |
| Sub-vastus | 9,671 (1.1) | 1,016 (1.1) | 546 (0.9) | 470 (1.3) |
| Other | 16,576 (1.9) | 2,212 (2.3) | 1,445 (2.4) | 767 (2.2) |
| **Fixation (%)** | | | | |
| Cemented | 814,854 (95.1) | 92,896 (96.2) | 59,009 (96.7) | 33,887 (95.5) |
| Uncemented | 34,999 (4.1) | 2,820 (2.9) | 1,681 (2.8) | 1,139 (3.2) |
| Hybrid | 6,684 (0.8) | 828 (0.9) | 358 (0.6) | 470 (1.3) |
| **Constraint (%)** | | | | |
| Constrained condylar | 6,906 (0.8) | 455 (0.5) | 280 (0.5) | 175 (0.5) |
| Monobloc poly tibia | 12,902 (1.5) | 2,664 (2.8) | 2,151 (3.5) | 513 (1.5) |

*(Continued)*

**Table 1.** (Continued)

| Variable | Surgeon grade and supervision (n=953,081) | | | |
| --- | --- | --- | --- | --- |
| | Consultant (*n*=856,537) | Trainee (overall) (*n*=96,544) | Trainee supervised by scrubbed consultant (*n*=61,048) | Trainee not supervised by scrubbed consultant (*n*=35,496) |
| Posterior stabilised, fixed | 196,523 (22.9) | 20,635 (21.4) | 13,197 (21.6) | 7,438 (21.0) |
| Posterior stabilised, mobile | 10,007 (1.2) | 943 (1.0) | 670 (1.1) | 273 (0.8) |
| Preassembled/hinged/linked | 994 (0.1) | 80 (0.1) | 61 (0.1) | 19 (0.1) |
| Unconstrained, fixed | 579,257 (67.6) | 68,013 (70.5) | 41,960 (68.7) | 26,053 (73.4) |
| Unconstrained, mobile | 49,948 (5.8) | 3,754 (3.9) | 2,729 (4.5) | 1,025 (2.9) |
| **Patellar resurfacing (%)** | | | | |
| Patella resurfaced | 333,035 (38.9) | 33,309 (34.5) | 21,037 (34.5) | 12,272 (34.6) |
| **Funding source (%)** | | | | |
| NHS | 749,953 (87.6) | 96,432 (99.9) | 60,979 (99.9) | 35,453 (99.9) |
| Private | 106,584 (12.4) | 112 (0.1) | 69 (0.1) | 43 (0.1) |
| **Year of operation (%)** | | | | |
| 2003–2011 | 301,105 (35.2) | 41,112 (42.6) | 21,949 (36.0) | 19,163 (54.0) |
| 2012–2019 | 555,432 (64.8) | 55,432 (57.4) | 39,099 (64.0) | 16,333 (46.0) |

ASA, American society of anaesthesiologists; NHS, national health service; IMD, index of multiple deprivation; BMI, body mass index. Data are n (%) or mean (SD); denoted where applicable.

**Table 2. The unadjusted cumulative probability of all-cause revision of TKRs according to surgeon grade (exposure A) and supervision (exposure B).**

| Follow-up (years) | ODEP A* (%) | Consultant | | | Trainee (overall) | | | Trainee supervised by a scrubbed consultant | | | Trainee not supervised by a scrubbed consultant | | |
| --- | --- | --- | --- | --- | --- | --- | --- | --- | --- | --- | --- | --- | --- |
| | | Number at risk* | Number of revisions | % Failure [95% CI] | Number at risk* | Number of revisions | % Failure [95% CI] | Number at risk* | Number of revisions | % Failure [95% CI] | Number at risk* | Number of revisions | % Failure [95% CI] |
| 1 | N/A | 856,537 | 3,460 | 0.42 [0.41, 0.44] | 96,544 | 434 | 0.47 [0.42, 0.51] | 61,048 | 287 | 0.49 [0.44, 0.55] | 35,496 | 147 | 0.42 [0.36, 0.50] |
| 3 | 3.5 | 689,676 | 7,756 | 1.53 [1.51, 1.56] | 81,169 | 993 | 1.67 [1.59, 1.76] | 49,366 | 598 | 1.68 [1.58, 1.80] | 31,803 | 395 | 1.65 [1.52, 1.79] |
| 5 | 4.0 | 517,132 | 3,580 | 2.21 [2.17, 2.24] | 63,659 | 430 | 2.33 [2.22, 2.44] | 37,018 | 268 | 2.39 [2.25, 2.53] | 26,641 | 162 | 2.24 [2.08, 2.41] |
| 7 | 4.5 | 365,234 | 1,892 | 2.70 [2.66, 2.75] | 47,189 | 249 | 2.84 [2.72, 2.96] | 26,136 | 151 | 2.95 [2.79, 3.11] | 21,053 | 98 | 2.69 [2.51, 2.88] |
| 10 | 5.0 | 188,465 | 1,602 | 3.42 [3.36, 3.47] | 25,143 | 174 | 3.34 [3.29, 3.59] | 13,038 | 92 | 3.54 [3.34, 3.75] | 12,105 | 82 | 3.29 [3.07, 3.52] |
| 13 | 6.0 | 64,200 | 701 | 4.23 [4.14, 4.31] | 8,922 | 103 | 4.25 [4.03, 4.48] | 4,609 | 55 | 4.44 [4.13, 4.78] | 4,313 | 48 | 4.02 [3.71, 4.34] |
| 15 | 6.5 | 24,663 | 153 | 4.79 [4.67, 4.92] | 2,922 | 17 | 4.75 [4.43, 5.10] | 1,597 | 6 | 4.74 [4.35, 5.16] | 1,325 | 11 | 4.76 [4.23, 5.35] |

Data are the number at risk, the number of revision events, the unadjusted cumulative probability of failure and the 95% confidence interval (CI). ODEP A* benchmark for comparison.

*Number at risk at the beginning of interval.

ODEP, Orthopaedic Device Evaluation Panel.

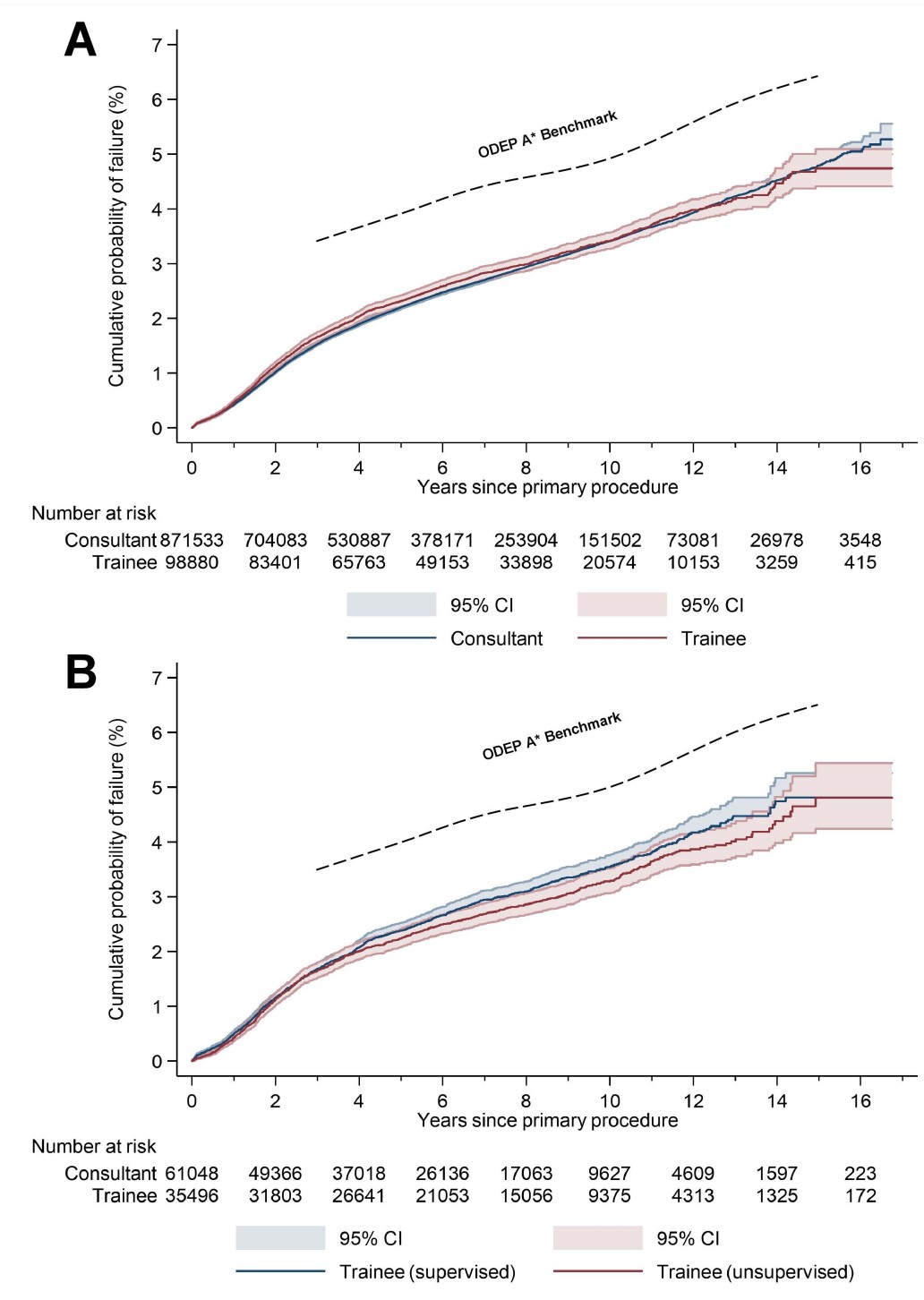

**Fig 2. Kaplan–Meier plots (one minus survival) demonstrating the cumulative probability of TKR failure (i.e., all-cause revision) according to surgeon grade (A) and supervision (B).**

An unadjusted sensitivity analysis demonstrates that surgeon grade has a relatively minor influence on implant survival compared to other operative factors, such as fixation, patellar resurfacing, and constraint (S3 Fig).

**All-cause revision: Flexible parametric survival modelling (FPM)**

Surgeon grade had a significant time-dependent association, and it is an oversimplification to report single numeric hazard ratio (HR). The results of this analysis, which compare trainee-performed TKRs to a baseline of consultant-performed TKRs, are therefore displayed as HR plots (Fig 3). HRs and CIs are presented in a tabular format in S5 Appendix.

A near-linear reduction in HR was observed in the unadjusted model (Model 1) between 1- and 4-years follow-up, from 1.12 (95% CI [1.05, 1.19]) at 1 year to 1.00 (95% CI [0.95, 1.06]) at 4 years, with a flattening of risk beyond this point to 0.89 (95% CI [0.81, 0.98]) at 16 years. Despite extensive adjustment for confounding factors, the marginal association between trainee-performed TKR and risk of revision in the first 4 years remains consistent. In the fully adjusted Model 4, the HR declines from 1.16 (95% CI [1.09, 1.23]) at 1 year to 1.05 (95% CI [0.99, 1.10]) at 4 years. In response to the peer review process, two sensitivity analyses were performed to: (i) include BMI as a confounding variable in Model 4, and (ii) to restrict the analysis to NHS-funded cases performed in NHS hospitals. The results are displayed in Fig 3 and demonstrate comparable results to those displayed in the HR plots for Models 1–4.

Further analysis was performed to compare the risk of revision of TKRs performed by trainees who were supervised by a scrubbed consultant to TKRs performed by trainees who were not supervised by a scrubbed consultant (exposure B). There was no evidence of an association between the level of scrubbed consultant supervision of trainee-performed TKRs and the risk of all-cause revision (Fig 4 and S5 Appendix).

An additional analysis was performed following further subcategorisation of cases according to the specific training grade of the operating surgeon (exposure C). There was no evidence of an association between F1-ST2 or fellow-performed TKR and the risk of all-cause revision, noting that only a small proportion of trainee procedures were performed by surgeons of these grades (F1-ST2: 1,104/96,544 [1.1%]; Fellow: 3,301/96,544 [3.4%]) (Fig 5 and S5 Appendix). The majority of trainee cases were performed by ST3-ST8 surgeons (92,139/96,544 [95.4%]). As such, the HR plot for this group resembles our analysis for trainees overall (Fig 3). Furthermore, there was no evidence of an association between the level of scrubbed consultant supervision of TKRs performed by ST3-ST8 surgeons and the risk of all-cause revision (Fig 6 and S5 Appendix). There were insufficient cases performed by F1-ST2 surgeons and fellows to repeat this analysis for these groups.

**Indication for revision: Flexible parametric survival modelling (FPM)**

The three most common indications for revision in this cohort were aseptic loosening/lysis ($n = 6,244$), infection ($n = 5,683$), and instability ($n = 4,361$). Fully adjusted models are displayed in Fig 7 and indicate marginal associations between trainee-performed TKR and early revision for aseptic loosening/lysis (up to 3 years), infection (up to 3 years), and progression of OA (up to 5 years). There was no evidence of an association between surgeon grade and revision for instability, or unexplained pain.

## Discussion

This study of nearly 1 million primary TKRs with over 16 years of follow-up provides insight into the association between surgeon grade, the supervision of trainees, and TKR survival. Our unadjusted KM analysis indicates that trainees achieve comparable outcomes to consultants, in terms of all-cause revision of TKRs. Trainees achieve implant survivorship within the ODEP A* threshold, which is an internationally recognised benchmark for the best performing implant components [23]. This supports the interpretation that trainees in England and Wales achieve safe and acceptable TKR implant survival. However, our adjusted FPM analyses identify areas for potential improvement in the outcomes of trainee-performed TKR.

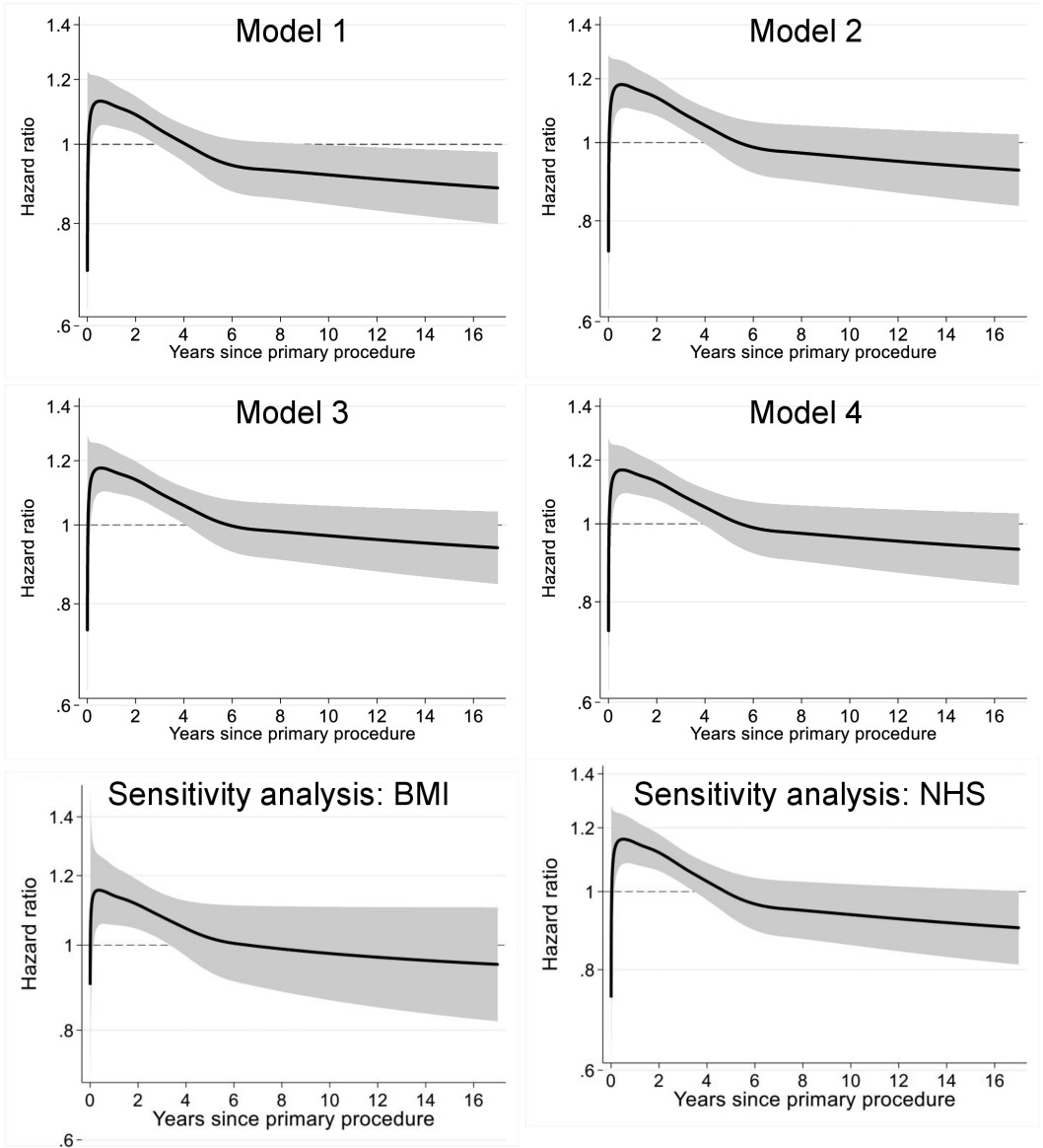

**Fig 3. Risk of all-cause revision of TKRs according to surgeon grade (exposure A).** HR plots for incrementally adjusted flexible parametric survival models (FPMs), which show the risk of revision for trainee-performed TKR compared to baseline of consultant-performed TKR. (Model 1) Unadjusted. (Model 2) Adjusted for patient factors. (Model 3) Adjusted for patient and operation factors. (Model 4) Adjusted for patient, operation and healthcare-setting factors. (Sensitivity Analysis: BMI) Adjusted for patient factors (including BMI), operation and healthcare setting factors (cases = 650,487). (Sensitivity Analysis: NHS) Adjusted for patient factors, operation and healthcare setting factors, but restricted to NHS-funded cases in NHS hospitals (cases = 636,670). The solid black line represents the HR, and the shaded grey area represents the 95% CI. Separation between the upper or lower 95% CI and the baseline of 1.00 (representing consultant-performed procedures) is suggestive of an association.

The use of FPM gives insight into the temporal variation in the risk of revision of TKRs. Our adjusted FPM analyses demonstrate that trainee-performed TKRs may be associated with a small increased risk of all-cause revision up to, but not exceeding, 4 years after the index procedure. Further analysis indicates that this association is not explained by the level of scrubbed consultant supervision and may be attributable to early revisions for aseptic loosening/lysis, infection, and progression of OA.

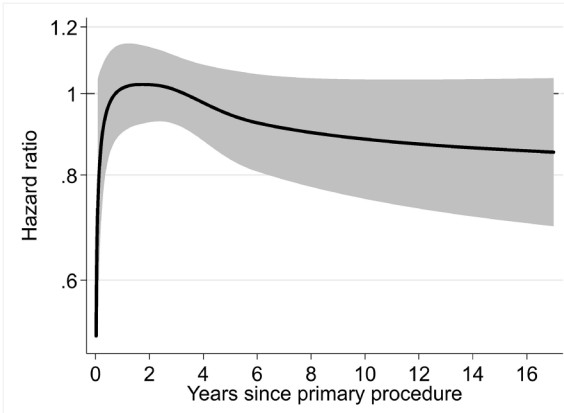

**Fig 4. Risk of all-cause revision of TKRs according to the level of supervision of trainees (exposure B).** HR plot for a fully adjusted flexible parametric survival model (Model 4). Represents TKRs performed by trainees supervised by a scrubbed consultant (baseline; cases = 61,048; revisions = 1,457) compared to TKRs performed by trainees not supervised by a scrubbed consultant (cases = 35,496; revision = 943). The solid black line represents the HR, and the shaded grey area represents the 95% CI. Separation between the upper or lower 95% CI and the baseline of 1.00 (representing TKRs performed by trainees supervised by a scrubbed consultant) is suggestive of an association.

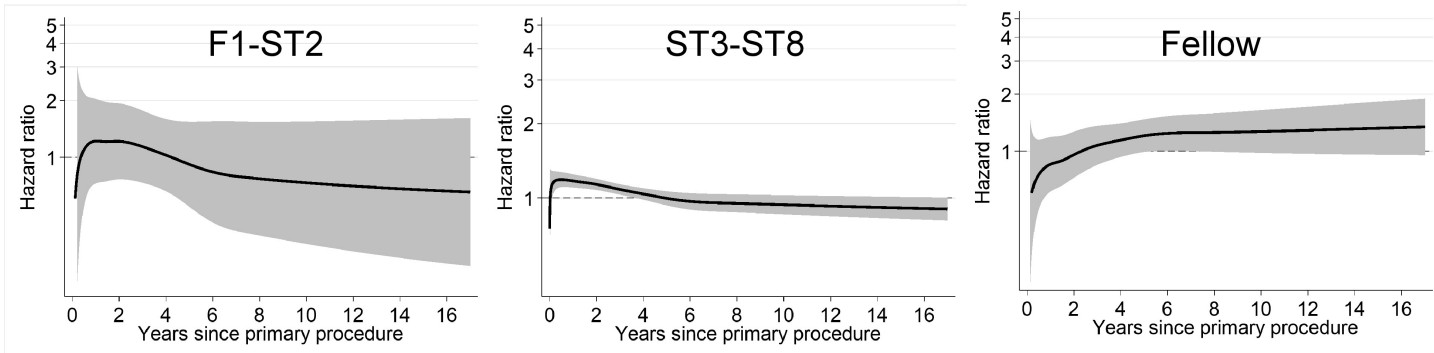

**Fig 5. Risk of all-cause revision of TKRs according to specific training grade (exposure C).** HR plot for fully adjusted flexible parametric survival models (Model 4). Risk of all-cause revision according to specific training grade (F1-ST2, ST3-ST8, Fellow) compared to baseline of consultant-performed TKR. F1-ST2: cases = 1,104; revisions = 29. ST3-ST8: cases = 92,139; revisions = 2,248. Fellow: cases = 3,301; revisions = 123. The solid black line represents the HR, and the shaded grey area represents the 95% CI. Separation between the upper or lower 95% CI and the baseline of 1.00 (representing consultant-performed procedures) is suggestive of an association.

The adjusted analysis shown in Fig 4 shows that we found no evidence of an association between the level of supervision of trainees and the risk of all-cause TKR revision. This might be because, in general, trainees who perform TKR without scrubbed consultant supervision are more experienced than trainees who are directly supervised by a scrubbed consultant. However, we were unable to quantify this in the current study. In practice, any observed difference in the risk of revision between consultant- and trainee-performed TKRs is very small, and surgeon grade has a relatively minor influence on implant survival compared to other operative factors, such as fixation, patellar resurfacing, and constraint (S3 Fig).Performing TKR without patellar resurfacing is associated with an increased risk of revision, compared to TKR with patellar resurfacing [24]. As such, patellar resurfacing is a cost-effective intervention, and National Institute for Health and Care Excellence (NICE) guidance suggests that it should be offered to patients undergoing an elective TKR [25]. Our results show that a lower proportion of trainee-performed TKRs included patellar resurfacing compared to

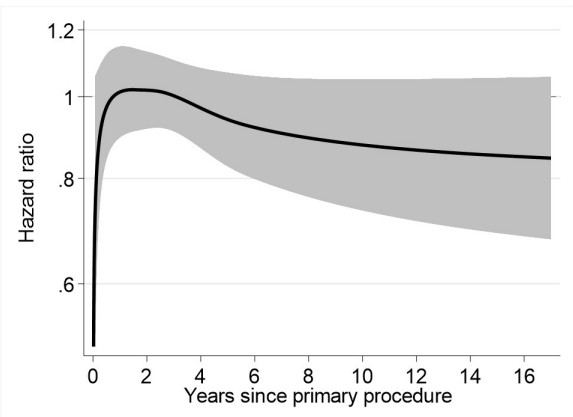

**Fig 6. Risk of all-cause revision of TKRs according to the level of supervision of ST3-ST8 trainees.** HR plot for a fully adjusted flexible parametric survival model (Model 4). Represents TKRs performed by ST3-ST8 trainees supervised by a scrubbed consultant (baseline; cases = 59,638; revisions = 1,407) compared to TKRs performed by ST3-ST8 trainees not supervised by a scrubbed consultant (cases = 32,501; revisions = 841). The solid black line represents the HR, and the shaded grey area represents the 95% CI. Separation between the upper or lower 95% CI and the baseline of 1.00 (representing TKRs performed by supervised ST3-ST8 trainees) is suggestive of an association.

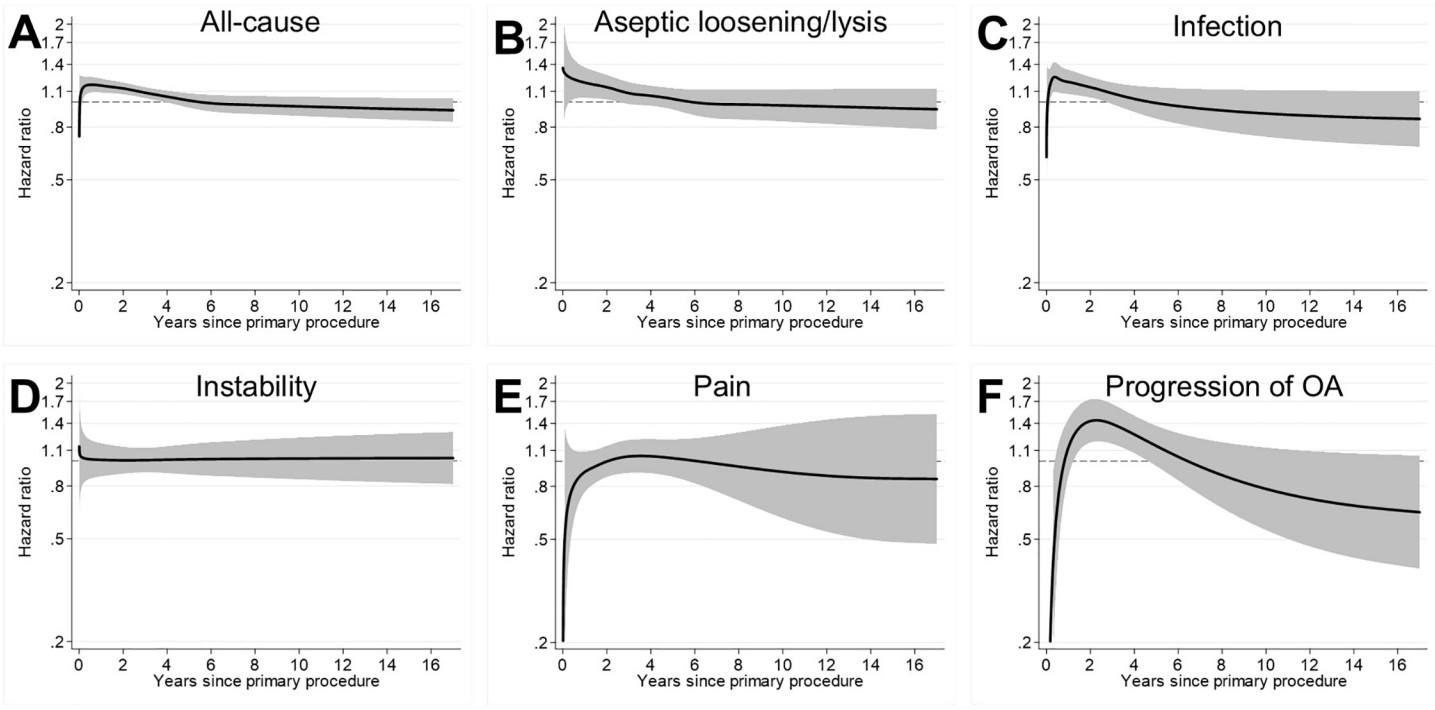

**Fig 7. The indication for TKR revision according to surgeon grade (exposure A).** HR plots for fully adjusted flexible parametric survival models (Model 4). Each plot (A–F) represents the risk of revision for trainee-performed TKR compared to a baseline of consultant-performed TKR. **(A)** All-cause revision [consultant revisions = 19,172; trainee revisions = 2,400]. **(B)** Aseptic loosening/lysis [consultant revisions = 5,563; trainee revisions = 681]. **(C)** Infection [consultant revisions = 5,025; trainee revisions = 658]. **(D)** Instability [consultant revisions = 3,917; trainee revisions = 444]. **(E)** Unexplained pain [consultant revisions = 2,853; trainee revisions = 327]. **(F)** Progression of OA [consultant revisions = 1,597; trainee revisions = 219]. N.B. Some cases were revised for more than one indication. The solid black line represents the HR, and the shaded grey area represents the 95% CI. Separation between the upper or lower 95% CI and the baseline of 1.00 (representing TKRs performed by supervised trainees) is suggestive of an association.

consultant-performed TKRs (34.5% versus 38.9%), which may explain the early risk of revision for progression of OA that was observed in the trainee cohort. We propose that trainers should consider patellar resurfacing, particularly when the trainee performs the TKR.

NZJR data suggest that mean operative duration is longer for trainee-performed TKR compared to consultant-performed TKR [7]. Large observational studies have identified an association between prolonged operative duration and prosthetic joint infection [26,27]. While causation cannot be attributed, the prolonged operative times previously observed in trainee-performed TKR, might explain the early risk of revision for infection observed in Fig 7C. Operative duration is not recorded in the NJR, which precluded analysis of this variable in the current study.

We analysed data for nearly 1 million knees recorded in the world's largest joint registry, which makes this the largest study of the association between surgeon grade and knee replacement outcomes [10,28]. We analysed data recorded in a mandatory national registry, which reduces sampling bias and improves the external validity and generalisability of our findings. Despite limiting our study period to predate the anomalous period of elective ortho-paedic practice during the COVID-19 pandemic, our findings are current and represent TKRs with up to 16.8 years of follow-up. OA was the only indication, which in addition to comprehensive confounding adjustment, reasonably accounts for measurable variations in case complexity and case-mix selection between the groups in the context of this observational study. We used FPM to account for non-proportionality by modelling the time-dependent effect of surgeon grade on the hazard function. This methodology gives unique insight into the temporal variation in the risk of revision according to surgeon grade.

Despite the strengths of this study, it has limitations. The data are observational, and patients were not randomly allocated to intervention groups; therefore, causation cannot be inferred. It is likely that more complex cases are prefer-entially performed by consultants. We have attempted to account for measurable differences in case complexity between the groups by adjusting for a comprehensive range of confounding variables and by restricting the analysis to cases performed for OA, which may help reduce confounding by indication. Furthermore, additional sensitivity analyses were conducted, including: (i) adjustment for BMI, (and ii) restriction to NHS-funded procedures in NHS hospitals, excluding procedures performed in private hospitals where trainees in England and Wales do not routinely operate. However, we acknowledge that residual confounding remains likely, and to our knowledge, no further adjustments are possible for fac-tors influencing case allocation to trainees. Nonetheless, our findings suggest that the current system for allocating cases to trainees in England and Wales is safe.

We also acknowledge the possibility that primary TKRs performed by trainees may be subject to a lower threshold for revision, particularly in the early postoperative period. This could reflect differing decision-making thresholds among supervising teams, though this could not be assessed within the available data.

Implant survival is an important objective measure of success. However, we did not consider other metrics that may be relevant when evaluating the success of a TKR, such as PROMS, or postoperative complications other than failure, as they are not currently reported by the NJR [29].

The binary term 'surgeon grade' does not capture variations in the level of experience between trainees. We have attempted to account for this by recategorising cases according to the specific training grade of the surgeon (exposure C: F1-ST2; ST3-ST8; fellow); however, this categorical variable has similar limitations. Similarly, supervision is recorded by the NJR as a binary variable according to the grade of the first assistant, which does not capture the wider spectrum of supervision, e.g., the supervision of trainees by unscrubbed consultants [30].

NJR data entry is audited on a rolling monthly basis to ensure data quality. Recent estimates suggest that over 96% of primary knee replacements and 91% of revision knee replacements are captured by the NJR [13]. However, there are likely to be limitations to the accuracy with which surgeon grade is recorded. For example, it is not currently possible to record more than one operating surgeon per case in the NJR, which does not account for procedures that have been part-performed by a trainee.

The results of our unadjusted KM analysis are consistent with the findings of previous observational studies, which are summarised in our recent systematic review and meta-analysis on this subject. Evidence synthesis was performed on 936 TKRs and suggested net implant survival estimates at 10 years of 96.2% (95% CI [94.0, 98.4]) for trainee-performed TKRs and 95.1% (95% CI [93.0, 97.2]) for consultant-performed TKRs [10]; comparable to the 10-year net failure estimates reported here (Table 2).

A NZJR study of 79,671 TKRs, which reported revision rates per 100 component years rather than net survival, found no significant difference in the revision rate of TKRs performed by consultants compared to TKRs performed by supervised and unsupervised trainees [7]. While these findings are generally consistent with the results of our unadjusted KM analysis, the high case numbers and statistical methods used in our study give additional insight into previously unknown associations between surgeon grade and the risk of revision following TKR. The same proportion of TKRs were performed by trainees in the NJR (10%) and the NZJR (10%).

The findings of our unadjusted KM analyses suggest that trainees achieve comparable all-cause implant survival to consultants. Trainees achieve implant survivorship within the ODEP A* threshold, which indicates safe and acceptable implant survival for TKRs performed by trainees in England and Wales. In general, trainers select cases of appropriate complexity for their trainees and permit trainees to operate without scrubbed supervision when they have reached a threshold of expertise that was not quantifiable in the current study. However, our adjusted analyses suggest that trainee-performed TKRs may be susceptible to an increased risk of early revision for aseptic loosening, infection, and progression of OA. We recommend that consultants and trainees should recognise and take appropriate measures to reduce the risk of failure from these indications. Trainees should ideally be supervised by a scrubbed consultant when performing TKR, particularly when junior and at critical stages of the procedure such as implant selection, balancing, fixation, and deciding whether or not to resurface the patella.

While our adjusted analyses identified small differences in early revision risk between consultant- and trainee-performed TKRs, the absolute differences were minimal. These marginal associations are unlikely to be of clinical significance, particularly when interpreted alongside our unadjusted analyses, which show that overall implant survival remains high and comparable across both groups. Taken together, these findings are reassuring and support the safety of trainee involvement in TKR, provided that appropriate case selection and supervision are in place.

This nationwide study of nearly 1 million TKRs demonstrates that trainees in England and Wales achieve safe and acceptable all-cause implant survival, with comparable outcomes to consultants. However, we have identified areas for potential further improvement in trainee outcomes. Trainee-performed TKRs may be susceptible to early revision for some specific causes. Careful patient selection, measures to prevent infection, and considered surgical decision-making may mitigate against the small early increased risk of revision for infection, aseptic loosening, and progression of OA in trainee-performed TKRs.

## Supporting information

**S1 Fig. Detailed study flow diagram showing sequential exclusions.**
(TIFF)

**S2 Fig. Detailed study flow diagram showing exclusion of missing data.**
(TIFF)

**S3 Fig. Kaplan–Meier plots (one minus survival) demonstrating the cumulative probability of failure according to the surgeon grade (exposure A); constraint; patellar resurfacing; and fixation.**
(TIFF)

**S1 Appendix. Process of accounting for changes in NJR operating surgeon grade categories.**
(DOCX)

**S2 Appendix. Schematic summary of surgical training in the UK.**
(DOCX)

**S3 Appendix. Model selection, construction and justification.**
(DOCX)

**S4 Appendix. Model specification summarising the exposures and confounding variables used in the analyses.**
(DOCX)

**S5 Appendix. Supplementary Results: Hazard ratio plots for** Figs 3–7 **presented in tabular format.**
(DOCX)

**S1 Note. Considerations Influencing All-Cause Revision.**
(DOCX)

**S1 RECORD Checklist. RECORD Checklist.**
(DOCX)

## Acknowledgments

We thank the patients and staff of all the hospitals who have contributed data to the National Joint Registry. We are grateful to the Healthcare Quality Improvement Partnership (HQIP), the National Joint Registry Steering Committee (NJRSC), and staff at the NJR Centre for facilitating this work. The views expressed in this publication are those of the authors and do not necessarily reflect those of the NHS, the NIHR, the UK Department of Health and Social Care, the NJRSC, or the HQIP.

## Author contributions

**Conceptualisation:** Timothy J. Fowler, Ashley W. Blom, Adrian Sayers, Michael R. Whitehouse.

**Data curation:** Timothy J. Fowler, Adrian Sayers.

**Formal analysis:** Timothy J. Fowler, Adrian Sayers.

**Funding acquisition:** Ashley W. Blom.

**Investigation:** Timothy J. Fowler, Ashley W. Blom, Adrian Sayers, Michael R. Whitehouse.

**Methodology:** Timothy J. Fowler, Ashley W. Blom, Adrian Sayers, Michael R. Whitehouse.

**Project administration:** Timothy J. Fowler, Adrian Sayers, Michael R. Whitehouse.

**Resources:** Timothy J. Fowler, Adrian Sayers, Michael R. Whitehouse.

**Software:** Timothy J. Fowler, Adrian Sayers.

**Supervision:** Ashley W. Blom, Adrian Sayers, Michael R. Whitehouse.

**Validation:** Nicholas R. Howells, Adrian Sayers, Michael R. Whitehouse.

**Visualisation:** Timothy J. Fowler, Nicholas R. Howells, Adrian Sayers, Michael R. Whitehouse.

**Writing – original draft:** Timothy J. Fowler, Michael R. Whitehouse.

**Writing – review & editing:** Timothy J. Fowler, Nicholas R. Howells, Ashley W. Blom, Adrian Sayers, Michael R. Whitehouse.

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
