## [Editor Report · Decision Letter 0]

26 Nov 2024

Dear Dr Fowler,

Thank you for submitting your manuscript entitled "Association between surgeon training grade and the risk of revision following total knee replacement: an analysis of National Joint Registry data" for consideration by PLOS Medicine.

Your manuscript has now been evaluated by the PLOS Medicine editorial staff as well as by an academic editor with relevant expertise and I am writing to let you know that we would like to send your submission out for external peer review.

Please re-submit your manuscript within two working days, i.e. by Nov 28 2024 11:59PM. Please do let us know if you need more time (ssunny@plos.org).

Kind regards,

Syba

Syba Sunny, MBBS, MRes, FRCPath

Associate Editor

PLOS Medicine

ssunny@plos.org

---

## [Decision Letter · Decision Letter 1]

11 Mar 2025

Dear Dr Fowler,

Many thanks for submitting your manuscript "Association between surgeon training grade and the risk of revision following total knee replacement: an analysis of National Joint Registry data" (PMEDICINE-D-24-04032R1) to PLOS Medicine. The paper has been reviewed by subject experts and a statistician; their comments are included below and can also be accessed here: [LINK]

After discussing the paper with the editorial team, I'm pleased to invite you to revise the paper in response to the reviewers' comments. We plan to send the revised paper to some or all of the original reviewers, and we cannot provide any guarantees at this stage regarding publication.

We ask that you submit your revision by Apr 01 2025 11:59PM. However, if this deadline is not feasible, please contact me by email, and we can discuss a suitable alternative.

Don't hesitate to contact me directly with any questions (atosun@plos.org).

Best regards,

Louise Gaynor-Brook, MBBS PhD

on behalf of

Alexandra Tosun, PhD

Associate Editor

PLOS Medicine

atosun@plos.org

Comments from the editors:

We agree with the comments from Reviewer 3 that additional analyses would enhance this manuscript, and we request that you include as many of the suggested analyses as possible in your revised version.

Comments from the reviewers:

Reviewer #1: This is an interesting registry based analysis of outcome of TKR compared to the level of surgical training concluding that there was little difference in revision rate between unsupervised and supervised trainees compared to consultant surgeons.

This data supplements previous studies showing similar results but has the value of being a very large data base.

The authors looked at reasons for revision and included this as one of the endpoints for comparison, which had not been studied before. They concluded that there was a small increase in revision rate for trainees, in particular they highlighted patellar resurfacing as an important difference between trainees and consultants and recommended all knees be resurfaced (based on their previous publication). Patellar resurfacing is a topic of debate, in particular resurfacing as a secondary procedure, which has often been criticized as not improving patient outcomes or relieving pain. The authors need to restrain their conclusions and offer a more reasoned approach to the small difference they have found as they offer no further data on outcomes of revision surgery in these cases ie did the revision actually improve the outcome. The threshold to revise TKR performed by a trainee may be lower and at least needs to be acknowledged by the authors.

Reviewer #2: "Association between surgeon training grade and the risk of revision following total knee replacement: an analysis of National Joint Registry data" examines possible associations between surgeon grade, supervision of trainees, and risk of revision following total knee replacement surgery (TKR). Data from the National Joint Registry (NJR) from England and Wales including over 950,000 cases, about 10% performed by trainees (about 63% under direct supervision by a consultant) were used in the analysis. It was found that TKR performed by trainees was associated with a small increased risk of early all-cause revision (independent of supervision status), from adjusted flexible parametric survival models (FPM). While the overall outcome was still within safe and acceptable bounds, areas for potential improvement were identified.

Strengths of the study include the scale of the dataset used. However, some issues might be considered:

1. In the Patients and data sources subsection, it is stated that inclusions were for indication of OA only. Was there any specific reason as to why other indications for TKR were not considered?

2. In the Data access and processing subsection, the inclusion process is described and illustrated in Figure 1 and S1/S2 Figure. However, "complete survival data" as detailed in S2 Figure does not appear to describe the completeness of temporal data. It might thus be clarified as to the requirements for follow-up data (i.e. if a patient does not have any visits recorded after the original surgery, is he considered as not requiring any revisions?)

3. In the Exposures subsection, it is stated that the secondary exposure was whether or not trainees were directly supervised by a scrubbed consultant during the procedure. It appears likely that there may be a supervision bias by trainee grade (i.e. F1 less likely to be unsupervised, compared to fellows). Moreover, the relevant figures (Figures 3 to 5) do not appear to compare revision hazard ratio by supervision status. This might be addressed.

4. Additionally, the choice of case assignment to trainees might be considered - would there be bias regarding the expected complexity of the case (i.e. easier/more routine cases would be assigned to trainees)? Related to this, Table 1 suggests that the proportion of cases handled by consultants versus trainees is significantly higher in 2012-2019 as compared to 2003-2011. This might be briefly discussed.

5. In the Outcomes of interest subsection, all-cause revision was stated as the primary outcome measure. The typical process of diagnosis/recommendation for such revision could be briefly discussed.

6. Figure 5 suggests higher revision for fellows, >4 years since primary procedure. This might be commented on - does this indicate a general delay in revisions compared to F1-ST8, or an overall increased rate of revisions over 16 years?

7. For Figure 5, it might be considered to standardize the scale of the y-axis for all charts (possibly increasing the height if required), for easier comparison; the corresponding chart for consultants could also be included.

8. In Figures 3 to 6, the 95% CIs for many of the charts appears similar at 0 years and 16 years, where it would be expected to be wider at 16 years due to a relative lack of data (from about 2003 only), as compared to at 0 years (where all data from 2003-2019 is relevant; this is reflected as expected in the Kaplan-Meier plots in Figure 2). These estimates might be confirmed.

9. In the Statistical analysis subsection, it is stated that records were censored at the end of 2019 or date of death, whichever is earliest. It might be clarified as to whether other factors (e.g. emigration) were considered.

10. In the Statistical analysis subsection, further details about FPM modelling were stated to be described in S2 Appendix. AIC & BIC details across various parameters such as degrees of freedom and knots might be included, if possible.

Reviewer #3: This is an interesting subject and the authors have attempted to answer the question of whether trainee performed TKR does as well as those performed by consultants. The methodology uses the NJR dataset, the limitations of which are well known and to a certain degree acknowledged in the paper. The aim of this paper should be to either change practice or confirm that current practice is acceptable. However, the analysis as reported is too simplistic to achieve either of these aims.

The weaknesses are considerable; the NJR dataset is non-verified for quality of what is input, there is no adequate case mix identification between groups, BMI is omitted due to missing values early in the NJR, revision for primary joint replacement may be a weak marker of "success" as opposed to PROMs or patient satisfaction measures, the disparity between consultant and trainee groups and the mix of settings (NHS v independent sector).

However, I think this is an important subject and it should be possible to obtain a more useful analysis.

I wonder whether the authors would consider a more in depth analysis? Even with only approximately 10% of TKRs being performed by trainees the number available to analyse are considerable. My suggestions would be:

Training has altered considerably since 2003. Trainees now perform considerably fewer procedures overall. Is there a difference in revision rate for trainee performed TKR over time?

Is there a difference in revision rate between trainees who have only TKRs registered as a trainee versus those trainees who have subsequent TKRs registered as a consultant?

Is there a difference in trainee revision rate as their numbers increase? e.g. does revision rate fall as experience increases? Is there a number of TKRs performed where the revision rate plateaus? Can you define the ranges of a learning curve?

Is the revision rate of unsupervised trainee TKRs affected by the number of supervised TKRs the trainee performed?

In terms of comparison to consultant performance, limiting the consultant group to NHS procedures in NHS hospitals would limit the confounders as you are then comparing within the same population (I assume just about all the trainee procedures are in NHS hospitals).

Reconsider not omitting BMI. I appreciate that there are many missing entries in the earlier years, but it is a good indicator of difficulty and there is an increased risk of infection in high BMI and infection is one of the main reasons for revision in the trainee group.

The FFD and ROM questions on the form allow some quantification of difficulty of the procedure. Does a FFD affect revision rate?

It may then be possible to either confirm current practice is safe, but it might also provide some insight into how the chance of a poor outcome may be minimised and training optimised.

---

* Please upload any figures associated with your paper as individual TIF or EPS files with 300dpi resolution at resubmission; please read our figure guidelines for more information on our requirements: http://journals.plos.org/plosmedicine/s/figures. While revising your submission, please upload your figure files to the PACE digital diagnostic tool, https://pacev2.apexcovantage.com/. PACE helps ensure that figures meet PLOS requirements. To use PACE, you must first register as a user. Then, login and navigate to the UPLOAD tab, where you will find detailed instructions on how to use the tool. If you encounter any issues or have any questions when using PACE, please email us at PLOSMedicine@plos.org.

FIGURES AND TABLES

SUPPLEMENTARY MATERIAL

REFERENCES

OBSERVATIONAL STUDIES

* Abstract: Please include the study design, population and setting, number of participants, years during which the study took place (enrolment and follow up), length of follow up, and main outcome measures.

* Please ensure that the study is reported according to the STROBE (or appropriate STOBE extension) guideline (available from: https://www.equator-network.org/reporting-guidelines/strobe) and include the completed STROBE (or STROBE extension) checklist as Supporting Information. Please add the following statement, or similar, to the Methods: "This study is reported as per the Strengthening the Reporting of Observational Studies in Epidemiology (STROBE) guideline (S1 Checklist)." When completing the checklist, please use section and paragraph numbers, rather than page numbers.

* For all observational studies, in the manuscript text, please indicate: (1) the specific hypotheses you intended to test, (2) the analytical methods by which you planned to test them, (3) the analyses you actually performed, and (4) when reported analyses differ from those that were planned, transparent explanations for differences that affect the reliability of the study's results. If a reported analysis was performed based on an interesting but unanticipated pattern in the data, please be clear that the analysis was data driven.

* Please state in the Methods section whether the study had a prospective protocol or analysis plan. If a prospective analysis plan (from your funding proposal, IRB or other ethics committee submission, study protocol, or other planning document written before analyzing the data) was used in designing the study, please include the relevant document(s) with your revised manuscript as a Supporting Information file to be published alongside your study and cite it in the Methods section. A legend for this file should be included at the end of your manuscript. If no such document exists, please make sure that the Methods section transparently describes when analyses were planned, and when/why any data-driven changes to analyses took place. Changes in the analysis, including those made in response to peer review comments, should be identified as such in the Methods section of the paper, with rationale.

---

## [Decision Letter · Decision Letter 2]

20 Jun 2025

Dear Dr. Fowler,

Thank you very much for re-submitting your manuscript "Association between surgeon training grade and the risk of revision following total knee replacement: an analysis of National Joint Registry data" (PMEDICINE-D-24-04032R2) for review by PLOS Medicine.

Thank you for your detailed response to the reviewers' and editors’ comments. I have discussed the paper with my colleagues, and it has also been seen again by all of the original reviewers. The changes made to the paper were satisfactory to the reviewers. As such, we intend to accept the paper for publication, pending your attention to the reviewers' and editors' comments below in a further revision. When submitting your revised paper, please once again include a detailed point-by-point response to the editorial comments.

The remaining issues that need to be addressed are listed at the end of this email. Any accompanying reviewer attachments can be seen via the link below. Please take these into account before resubmitting your manuscript: [LINK]

In revising the manuscript for further consideration here, please ensure you address the specific points made by each reviewer and the editors. In your rebuttal letter you should indicate your response to the reviewers' and editors' comments and the changes you have made in the manuscript. Please submit a clean version of the paper as the main article file. A version with changes marked must also be uploaded as a marked up manuscript file. Please also check the guidelines for revised papers at http://journals.plos.org/plosmedicine/s/revising-your-manuscript for any that apply to your paper.

We ask that you submit your revision within 1 week (Jun 27 2025). However, if this deadline is not feasible, please contact me by email, and we can discuss a suitable alternative.

Please do not hesitate to contact me directly with any questions (atosun@plos.org). If you reply directly to this message, please be sure to 'Reply All' so your message comes directly to my inbox.

We look forward to receiving the revised manuscript.

Sincerely,

Alexandra Tosun, PhD

Senior Editor 

PLOS Medicine

plosmedicine.org

Comments from Reviewers:

Reviewer #1: Thank you I am satisfied with your responses

Reviewer #2: We thank the authors for their detailed responses, and commitment to release related code upon publication. Nonetheless, regarding the diagnostic criteria for all-cause revision, some additional detail on such might be appropriate in the supplementary material. In particular, the effect of follow-up consultation frequency, and any cost/insurance/functional quality-of-life considerations (if relevant) might be discussed as to whether and to what extent they might affect a revision operation taking place.

Reviewer #3: My comments have been addressed adequately given the limitations of the data available in the NJR. The lack of ability to provide a better analysis that could influence future training and hence improve patient outcomes is disappointing. However, it is an important subject and the finding of adequate revision rates from trainee led TKRs is helpful. The discussion covers the limitations appropriately.

[LINK]

Requests from Editors:

GENERAL

* Please confirm that your title complies with to PLOS Medicine's style. Your title must be nondeclarative and not a question. It should begin with main concept if possible. "Effect of" should be used only if causality can be inferred, i.e., for an RCT. Please place the study design ("A randomized controlled trial," "A retrospective study," "A modelling study," etc.) in the subtitle (i.e., after a colon).

* Statistical reporting: Please revise throughout the manuscript, including tables and figures.

- Please report statistical information as follows to improve clarity for the reader ""22% (95% CI [13,28]; p</=)"".

- Please separate upper and lower bounds with commas instead of hyphens as the latter can be confused with reporting of negative values.

- Please repeat statistical definitions (HR, CI etc.) for each set of parentheses.

* Please ensure that all abbreviations are defined at first use throughout the text (including statistical abbreviations). Please also check figures and tables.

* Please ensure that tables and figures, including those in supplementary files, are appropriately referenced in the main text.

* Financial Disclosure: If available, please provide URL(s) of each funder website.

* Please check that any use of statistical terms (such as trend or significant) are supported by the data, and if not please remove them.

* You have stated that you are happy to make author-generated code for this paper available on publication. Please add a sentence to your data availability statement regarding any code used in the study, e.g. "The code used in the analysis is available from Github [URL] and archived in Zenodo [DOI link]" Please ensure that your code is shared in a way that follows best practice and facilitates reproducibility and reuse.

ABSTRACT

* Please confirm that your abstract complies with our requirements, including providing all the information relevant to this study type https://journals.plos.org/plosmedicine/s/submission-guidelines#loc-abstract

* Please ensure that all numbers presented in the abstract are present and identical to numbers presented in the main manuscript text.

* If the data is available, we think it would be helpful to mention the number of trainees and consultants you assessed.

* Given our international readership, we believe it would be useful to provide the average number of years of experience for trainees and consultants.

* Please quantify the main results (with 95% CIs and p values).

* Your study is observational and therefore causality cannot be inferred. Please remove language that implies causality, such as effect. Refer to associations instead. Please revise throughout the main text.

AUTHOR SUMMARY

* If you prefer, you may write the author summary from a first-person plural perspective, i.e., "we."

INTRODUCTION

* If you agree, we suggest defining the ‘specific indications’ in the last sentence.

METHODS AND RESULTS

* You stated that the study protocol was planned in detail prior to commencing the study, yet you also said that such a document does not exist. Please clarify. If the document exists and was used to design the study, include it as a Supporting Information file in your revised manuscript to be published alongside your study, and cite it in the Methods section. If the document does not exist, revise the statement in the main text for clarity.

* Do you have data on how many trainees and consultants performed TKRs? If so, we think it would be valuable to include these.

* We have noted that you switch between the descriptions “all-cause failure” and “all-cause revision” for the primary outcome. We recommend using the same term consistently.

* You discuss marginal differences between trainee and consultant outcomes multiple times. Given these small statistical differences, we think it’s important to discuss the clinical relevance of these findings. We have highlighted some of the relevant statements below:

- “There is a subtle divergence in the probability of failure between 1 and 4 years, with separation between the confidence intervals noted at the 3-year interval.”

- “Despite extensive adjustment for confounding factors, the marginal association between trainee-performed TKR and risk of revision in the first 4 years remains consistent.”

- Fully adjusted models are displayed in Figure 7 and indicate marginal associations between trainee-performed TKR and early revision for aseptic loosening/lysis (up to 3 years), infection (up to 3 years), and progression of OA (up to 5 years).

- “Surgeon grade had a significant time-dependent effect” – We feel this is an overstatement and the term ‘effect’ should only be used if causality can be inferred.

* Figure 3/4/5/6: Please provide a more specific title for the y-axis.

* Figure 1: Please define all abbreviations used in the figure.

* Figure 3: For the two graphs depicting the results of the sensitivity analyses, please include the numbers of cases included.

* Figure 4/6: Please include case numbers.

* Figure 7: Please ensure that the y-axis is identical for all figures to facilitate comparison.

DISCUSSION

* Please review your text for claims of novelty or primacy (e.g. 'for the first time') and remove this language.

* Pleas remove all subheadings.

General Editorial Requests

---

## [Editor Report · Decision Letter 3]

7 Jul 2025

Dear Dr. Fowler,

Thank you very much for re-submitting your manuscript "Association between surgeon training grade and the risk of revision following total knee replacement: an analysis of National Joint Registry data" (PMEDICINE-D-24-04032R3) for review by PLOS Medicine.

Thank you for your detailed response to the reviewers' and editor’s comments. There are a few minor editorial issues that need to be addressed before we can accept the manuscript for publication. When submitting your revised paper, please once again include a detailed point-by-point response to the editorial comments. Please revise the paper accordingly, and submit the final revision by July 10. The remaining issues that need to be addressed are listed at the end of this email.

In revising the manuscript for further consideration here, please ensure you address the specific points made by the editors. In your rebuttal letter you should indicate your response to the editors' comments and the changes you have made in the manuscript. Please submit a clean version of the paper as the main article file. A version with changes marked must also be uploaded as a marked up manuscript file. Please also check the guidelines for revised papers at http://journals.plos.org/plosmedicine/s/revising-your-manuscript for any that apply to your paper.

A reminder that when your manuscript is accepted, an uncorrected proof of your manuscript will be published online ahead of the final version, unless you've already opted out via the online submission form. If, for any reason, you do not want an earlier version of your manuscript published online or are unsure if you have already indicated as such, please let the journal staff know immediately at plosmedicine@plos.org.

Please do not hesitate to contact me directly with any questions (atosun@plos.org). If you reply directly to this message, please be sure to 'Reply All' so your message comes directly to our inbox.

We look forward to receiving the revised manuscript.

Sincerely,

Alexandra Tosun, PhD

Senior Editor

PLOS Medicine

plosmedicine.org

Requests from Editors:

* In the Abstract Background, please explain “revision”. For example: Revision surgery is any subsequent surgery performed to correct or improve the results of an earlier procedure.

* In the Abstract, please include the important dependent variables that are adjusted for in the analyses (briefly explain Model 1, 2, 3 and 4).

* Abstract: Thank you for your response to our request for the average number of years of experience for trainees and consultants. Upon further consideration, we suggest removing the text.

* While we understand that your analysis presents data in a time-dependent nature, we are convinced that the inclusion of numerical results is essential for accurate reporting. Please note that we have some leeway regarding Abstract length for the sake of reporting transparency. We have outlined two suggestions below:

- Suggestion #1: “Trainees achieved comparable outcomes to consultants in terms of the unadjusted cumulative probability of all-cause revision (e.g. 5 years of follow-up: consultant % Failure 2.21 (95% CI [2.17, 2.24]) versus trainee (overall) % Failure 2.33 (95% CI [2.22, 2.44])).”

- Suggestion #2: “Adjusted FPM analysis indicated evidence of an association between trainee-performed TKR and a small increased risk of early all cause revision up to, but not exceeding, 4 years follow-up (1 year: HR 1.12 (95% CI [1.05, 1.19]), 4 years: HR 1.00 (95% CI [0.95, 1.06]), 16 years: HR 0.89 (95% CI [0.81, 0.98])).”

* For Figure 3-7, please provide the hazard ratios and CIs in tabular format as a Supporting Information file.

* “Surgeon grade had a significant time-dependent effect” (l.527 track changes) – Please note that although you said you adjusted the statement, it remains unrevised.

* S3 Figure: In the Kaplan-Meier curve(s) please provide the number at risk for each time interval.

---

## [Editor Report · Decision Letter 4]

16 Jul 2025

Dear Dr Fowler, 

On behalf of my colleagues and the Guest Academic Editor, Gary Hooper, I am pleased to inform you that we have agreed to publish your manuscript "Association between surgeon training grade and the risk of revision following total knee replacement: An analysis of National Joint Registry data" (PMEDICINE-D-24-04032R4) in PLOS Medicine.

I appreciate your thorough responses to the reviewers' and editors' comments throughout the editorial process. We look forward to publishing your manuscript, and editorially there is only one remaining point that should be addressed prior to publication. We will carefully check whether the request has been addressed. If you have any questions or concerns regarding this final request, please feel free to contact me at atosun@plos.org.

Please see below the minor point that we request you respond to:

* Abstract: We suggest also including the second suggestion with numerical results. Editorial suggestion: “Adjusted FPM analysis indicated evidence of an association between trainee-performed TKR and a small increased risk of early all cause revision up to, but not exceeding, 4 years follow-up (1 year: HR 1.12 (95% CI [1.05, 1.19]), 4 years: HR 1.00 (95% CI [0.95, 1.06]), 16 years: HR 0.89 (95% CI [0.81, 0.98])).”

Before your manuscript can be formally accepted you will need to complete some formatting changes, which you will receive in a follow up email (including the editorial point above). Please be aware that it may take several days for you to receive this email; during this time no action is required by you. Once you have received these formatting requests, please note that your manuscript will not be scheduled for publication until you have made the required changes.

PRESS

Sincerely, 

Alexandra Tosun, PhD 

Senior Editor 

PLOS Medicine